# Social Farming: An Inclusive Environment Conducive to Participant Personal Growth

**Roberta Moruzzo** [1], **Francesco Di Iacovo** [1], **Alessandra Funghi** [2], **Paola Scarpellini** [3], **Salomon Espinosa Diaz** [1] **and Francesco Riccioli** [1,*]

[1] Department of Veterinary Science—Rural Economics Section, University of Pisa, Viale Delle Piagge 2, 56124 Pisa, Italy; roberta.moruzzo@unipi.it (R.M.); francesco.diiacovo@unipi.it (F.D.I.); saloespinosadiaz@gmail.com (S.E.D.)

[2] Self-Employed Agronomist, Via 1 Maggio 35, Pontedera, 56025 Pisa, Italy; alessandra.funghi@gmail.com

[3] Department of Economics and Business Sciences, University of Florence, Via Delle Pandette 9, 50127 Florence, Italy; paola.scarpellini@unifi.it

[*] Correspondence: francesco.riccioli@unipi.it

**Abstract:** Social farming can ameliorate the everyday life of people engaged in farming activities, including perceived changes in mood or behavior. It can also be therapeutic, as it can address a range of public health and service provision issues. This paper presents the findings of an Italian project that explored the impact of social farming on the well-being of the participants and their ability to perform certain tasks linked to agricultural activities. In addition, this paper tries to evaluate how the organization of the network system around the participants helps them to improve their relational capabilities. Participant observations were made in class rooms and farms where the participants carried out their agricultural activities. Such observations focused on the way in which participants and other subjects (i.e., tutors and training staff) inside the network system interacted. A number of in-depth interviews were carried out with tutors and trainers in order to understand if the participants would play a relevant role in social farming activities and what that role would be.

**Keywords:** social farming; rural welfare; community network; social inclusion; care farming

## 1. Introduction

In the literature, there is a wide range of activities and practices that have in common the use of natural elements to maintain and promote physical, mental and social well-being (Hassink and Van Dijk 2006; Hine et al. 2008; Di Iacovo and O'Connor 2009; Sempik 2010). There are several ways to define this complex and heterogeneous phenomenon. Green care is an inclusive and umbrella term that includes a broad variety of interventions such as nature-based rehabilitation, care farming, social farming, therapeutic horticulture, and animal-assisted intervention (Garcia-Llorente et al. 2018). These concepts are (1) used as synonyms sometimes, (2) they are supported by different academic backgrounds and theories and, (3) they have a different level of development and representation that varies according to the country (for example, green care in the Netherlands, therapeutic horticulture or care farming in the UK, farm animal-assisted intervention in Norway, social farming in Italy and Spain). In particular, in many research publications there are relevant interconnections or similarities between two different concepts (e.g., care farming—CF and social farming—SF), which explains why the terms are often used interchangeably (Guirado et al. 2017). According to (Dessein et al. 2013), the concepts of SF and CF, are used to describe the complexity and territorial specificities of a phenomenon that links agricultural work to the provision of health and human services. When it comes to these two concepts the objectives are more related to conducting occupational activities and achieving employment goals

at real production farms (Hassink et al. 2016). However, the therapeutic purpose of SF is not so explicit (Di Iacovo et al. 2017). SF not only promotes improved quality of life but also contributes to the creation of new local development strategies (European Network for Rural Development 2010), forging alliances between agriculture and social care, generating new models of care and welfare, and promoting the creation of new personal services in rural areas (Di Iacovo et al. 2014).

In spite of the increment in the number of research publications about practices related to the provision of health and human services (Elings and Hassink 2008; Dessein et al. 2013; MIND (Mental Health Association); Hassink et al. 2014; Hemingway et al. 2016), this topic is still poorly investigated and understood (Bombach et al. 2015; Leck et al. 2014). Harnessing the benefits of nature on human health through interventions that include "ecotherapy" (Burls and Caan 2005; MIND (Mental Health Association)) and "therapeutic horticulture" (Sempik 2010) is being increasingly viewed as an effective way to improve mental health levels in many parts of the world. However, the evaluation of other alternatives like social farming and care farming is still limited (Gagliardi et al. 2019; Garcia-Llorente et al. 2018). In Europe in particular, social farming has frequently been framed in the field of health sciences incorporating some research on its therapeutic effects and the impact it has on some indicators of health and well-being. Some researchers have even started to compare the effectiveness of social farming or care farming to that of other therapeutic processes (Eriksson et al. 2011).

The perceived benefits of care farms are improved physical, mental, and social well-being (Hassink et al. 2010). According to some authors (Elings and Hassink 2008; Hemingway et al. 2016; Hine et al. 2008), some of these mental health benefits consist of an improvement in self-esteem, well-being and mood and some social benefits are independence, formation of work habits and the development of personal responsibility and social skills. These authors have found that care farms offer a homely, supportive environment where people can experience nature and sustainable food production. They perceived care farms to be a place that provides an inclusive environment conducive to their clients' personal growth, which enables them to connect with themselves, others and nature and to develop autonomy. Another benefit is that people at care farms have the opportunity to learn about themselves and nature, but in general terms all these benefits can differ by type of participant.

Some studies on participants with clinically diagnosed mental illnesses have demonstrated improvements in mental health scores after spending time on a care farm (Sempik et al. 2014). Farms provide a platform where people can find meaning to their lives as well as relationships that grow and become an important part of their social wellbeing (Hassink et al. 2010; Iancu et al. 2014; Kogstad et al. 2014; Leck et al. 2015).

Some authors also examined the effect that being in contact with nature has on people suffering with depression (Gonzalez et al. 2010; Pedersen et al. 2012). They found a statistically significant fall in depression scores in a group of patients with moderate to severe depression attending a therapeutic garden project. An evaluation of a garden project for ex-servicemen with post-traumatic stress disorder reported that both clinical staff and patients viewed the project as having "positive therapeutic benefits" (Atkinson 2010).

Other studies have also found that social farming can provide care to elderly people with mild or severe signs of dementia (De Bruin et al. 2009; De Bruin et al. 2012; Schols and van der Schriek-van Meel 2006). At the same time these studies have observed a positive level of appreciation from the participants regarding their personal relationships with farmers, the informal—not clinical—context of the activities, and the opportunity to perform a variety of useful activities (De Bruin et al. 2015; Hassink et al. 2010). In a study by De Boer et al. (2016), dementia patients on a social farm were found to be more likely to participate in outdoor activities and green exercise and to engage more in social interactions than those in nursing homes. Gagliardi et al. (2019) conducted a study in order to evaluate a one-year social farming programme conducted between 2014 and 2015, which included horticultural and occupational activities on six agricultural farms for older people in good health condition. Data analysis revealed significant improvements in both social relationships and overall

occupational engagement at the end of the programme, with a significant increment in the frequency of contact with friends or relatives as well as in the number of activities performed by the participants.

## 2. Building Social Farming in the Italian Welfare System: The "We Too" Project

Generally speaking, the social welfare service aims to reinforce sustainable social participation (Lee et al. 2019; Lyngstad 2015). In Italy, social farming includes a wide range of practices and activities supporting a new welfare system idea (Ismea 2017) and it is recognized by law (L141/2015) as an activity provided by farms in connection with local public health/social authorities in order to provide co-therapy with plants and animals, social inclusion, training, job creation, civil services for diverse target groups and education on farming for people in difficulty.

The Law 141/2015 defines the scope of social farming in four specific typologies:

Social and working inclusion of people belonging to the weakest sectors, acknowledged by local and regional welfare bodies and the working and social inclusion of disadvantaged and disabled people;

Social, socio-sanitary, rehabilitative, therapeutic, training and educational services for families, seniors, disadvantaged and disabled people;

Social activities to support local communities, implementing the use of material and immaterial agricultural resources to provide services useful for everyday life, as well as promoting, supporting and achieving actions of social and occupational inclusion, recreation and education;

Educational activities addressed to vulnerable people.

This Italian law was the result of a bottom-up process starting from project-holders and progressively involving institutions and political parties both at a regional and eventually, at a national level.

The Italian model of social farming, unlike that of other European countries, overcomes state/market logic and ensures the co-production of economic and social values of public and private nature as well as values related to food and health; it is based on the proactive behaviour of diverse actors involved in new local community alliances. For these reasons, Italian social farming represents a practical translation of the new principles of European Welfare into implemented practices (Di Iacovo 2009; Dessein et al. 2013).

In North European countries, care farming practices are paid to agricultural farms accredited to the national health system as activities of economic diversification for farms and diversification of personal services offered by the national welfare. In France and Germany, similarly, they are supported by the health service and entrusted to actors in the third sector in structures with a high socio-sanitary prevalence (Di Iacovo 2009).

In the Italian context, things are completely different and, in many aspects, much more consistent with the debate on rural welfare and the need for a re-generative welfare model based on public–private collaboration, the creation of networks for the management of the services offered (Giarè et al. 2018) and the identification of shared solutions to complex problems (Berti 2012; Dessein et al. 2013; Di Iacovo et al. 2018, 2014; Di Iacovo and O'Connor 2009; Genova n.d.; Hassink et al. 2010; Di Iacovo et al. 2018).

As argued by Di Iacovo et al. (2017), the Italian law outlines a space to share experiences about the development of a program that integrates policies among farms, services and local institutions. This way, social farming goes beyond the multifunctional role of agriculture and impacts on the communities of rural and peri-urban territories with an existing offer of services, affecting the welfare system as a whole. In particular, agriculture emerges as a social innovation practice useful for strengthening the

provision of services in rural areas, but also for revitalizing networks and relationships at individual and community levels.

The project "We too" (2016–2018), developed in the province of Pisa by the Local Health authority and supported by the European Social Fund (ESF), is an example of social farming initiatives in Italy. It promotes the integration of disabled and/or mentally ill people into the working world, providing them with a set of qualifications, knowledge, skills and competencies that can be used for this purpose.

The strategy defines "individualized implementation plans" aimed to identify the subject's residual capacities, providing more appropriate training and/or job placement pathway and creating a support system that sustains, coordinates and constantly monitors the development of such individualized plans.

The design process has been developed according to the following operational directives in which different actors were involved: selection and orientation of the participants; pre-training class room activities; a stage on farms. The selection and orientation phase was conducted by a special multidisciplinary commission in the presence of social services that organized individual and group meetings with a counsellor to define the attitudes and the main predispositions of each participant. In this phase, the role of the counsellor was to allow participants to attend the most relevant class room/farm for their own capabilities (ex-ante orientation).

Participants could follow three different paths:

Path 1 (pre-training class room): participants were assigned only to a class room (which could be a social cooperative or an association inside the project) without the possibility of continuing the path onto a farm due to their limited capacities,

Path 2 (pre-training class room + stage on a farm): participants for whom a class room placement was deemed useful were assigned to this path. At the end of the class room period (in general, three months), the participant continued the path (for another six months) on an agricultural farm of the territory.

Path 3 (stage on a farm): some participants were assigned directly to an agricultural farm (without going to a class room), since their condition was considered to be adequate to do so.

The orientation phase, carried out by a counsellor, also continued during the pre-training class room/stage on farms, and counted on the presence of representatives of the agricultural associations (CIA and Coldiretti) in order to support and manage moments of possible difficulty or crisis of the participants (on-going orientation) and to ensure the provision (for the participants of Path 2) of the most complete information for the realization of farm scouting and personalized matching (ex-post orientation).

During the pre-training class room activities and the farm stage, two individuals (a tutor and a farm/class room training staff member) were assigned to each participant:

Farm or class room training staff guaranteed agricultural technical training for the participants;

Tutors were primarily responsible for supporting participants during the pre-training class room and during the farm stage, trying to identify the levels of participant capacity for autonomy and self-esteem. The tutors were always present in the class rooms during the path, but their presence at the farm was less constant.

The purpose of this paper was to analyse the project described before as a case study, examining the organization of the strategies adopted and measuring the impact that these activities had on the everyday lives of the participants, including any perceived changes in mood or behaviour. Careful consideration is also given to whether participants feel more confident about themselves or not, their ability in agricultural activities and the evaluation of how the organization of the network system around the participants helped them to improve their self-assessment capacity and their relational skills. Based on the described analysis this study will answer the following questions: Can the

participants acquire greater awareness about their own skills through their participation in these paths? Are improvements more evident in psycho-attitudinal skills or in technical skills? What are the main strengths and weaknesses of these types of strategies?

## 3. Methodology

### 3.1. Study Design

The study adopted a "case study" approach. This approach represents an intensive analysis of a single unit aiming to generalize achievements across a larger set of units (Gerring 2004; Pacho 2015). This particular approach was chosen since it allowed the experiences of people engaged in social farming activities to be explored. The principal means for data collection used in this study were participant observation made by the tutors and staff present in the class rooms and farms and in-depth interviews made by the authors.

Participant observation has been used in a variety of disciplines as a tool for collecting data about people, cultures, and processes in qualitative research (Kawulich 2005). (Taylor et al. 2016) define participant observation as a period of intense social interaction between the researcher and the subjects, during which data are unobtrusively and systematically collected. The process enables researchers to learn about the activities of the people under study in their natural setting, through observation and participation in those activities. The researcher attempts to get as close as possible to the group that he or she is studying to best understand it; observations are complemented with field notes (Hong and Duff 2002; Kawulich 2005).

Boyce and Neale (2006) identify in-depth interviews as a qualitative research technique that involves conducting intensive individual interviews with a small number of respondents to explore their perspectives on a particular idea, program, or situation. For example, it is possible to ask respondents associated with a program about their experiences and expectations related to the program, the thoughts they have concerning program operations, processes, outcomes and about any changes they perceive in themselves as a result of their involvement in the program. The questions asked during in-depth interviews need to be worded in a way that stimulates respondents to elaborate their answers on the topic, rather than simply answering "yes" or "no". As opposed to closed questions, open questions do not present the respondent with a list of possible answers from which to choose (Bryman 2008). This gives respondents the freedom to answer the questions using their own words and allows the interviewer to deeply explore the respondent's feelings and perspectives on a subject.

### 3.2. Data Collection

In the project "We too", at the end of the selection phase, 23 participants in total were recruited from different social services: 17 from mental health services (Path 2), 5 from disability services (Path 3) and one with a double diagnosis (Path 1). In order to analyze the impact of social farming on the participants, we decided to focus our analysis only on the participants who followed Path 2, which received a more complete training by attending both the class room and farm stage levels: a total of 9 users were assessed in our analysis.

This approach allowed the comparison of data collected (for the same participant during both the pre-training class room and the farm stage) as well as the analysis of improvements/worsening of the participants experience during the social farming activities. Generalities about participants and cultivation system of farms used for the stages are showed in Table 1.

For this reason, the fieldwork phase of this study comprised two rounds of data collection. The first one took place during the period in the class room; the second round took place approximately three months later, during the period at the farm. This particular research strategy was designed to track participant progress during the period of the social farming program.

**Table 1.** Participants generalities.

| ID | Age | Gender | Cultivation System |
|----|-----|--------|--------------------|
| 1 | 47 | male | vegetables |
| 2 | 33 | male | worms for feed |
| 3 | 44 | male | vegetables |
| 4 | 38 | male | vegetables |
| 5 | 52 | male | olive groves, farmhouse |
| 6 | 40 | male | olive groves, vineyards |
| 7 | 39 | female | vegetables |
| 8 | 52 | male | vegetables |
| 9 | 54 | male | vegetables |

Participant observations (made in pre-training class rooms and farms) focused on the way in which participants and other subjects inside the organization interacted, what they did and which roles they took during the interaction. During fieldwork the authors visited the class rooms and farms and the tutor and the farm staff (who have a better understanding of the context and phenomenon under study) collected extensive observational field notes on participant behavior, thoughts, feelings, and actions. Two separate forms, developed from an evaluation tool used by social services in the province of Pisa (Marini 2016) and linked to the Psychological General Well-Being Index (PGWBI; Dupuy 1984), were used for the collection of the data: the psycho-attitudinal skills assessment form; and the technical skills assessment form.

The psycho-attitudinal skills assessment form, compiled by the tutor in collaboration with the farm/classroom training staff, was a data collection form created specifically for each area of personal attributes (e.g., basic skills, psychosocial skills, and individual competencies). The indicators that were scored are described in Table 2. Respondents could answer by giving a score using a five-point Likert scale: ranging from totally disagree (1), disagree (2), neutral (3), agree (4) and totally agree (5).

**Table 2.** Indicators of psycho-attitudinal skills.

| Indicator | Skill |
|-----------|-------|
| 1 | The individual respects the rules |
| 2 | The individual respects and understands roles/responsibilities |
| 3 | The individual shows lucidity and concentration |
| 4 | The individual is motivated and interested |
| 5 | The individual is able to use tools with property and responsibility |
| 6 | The individual checks results and is able to solve problems |
| 7 | The individual is reliable |
| 8 | The individual shows productivity |
| 9 | The individual has personal care |
| 10 | The individual socializes and relates with the working group, recognizing each one's roles |
| 11 | The individual communicates clearly and directly |
| 12 | The individual performs different tasks with quality and flexibility |
| 13 | The individual has organizational skills |
| 14 | The individual shows good willingness to receive training |
| 15 | The individual has good autonomy and initiative at work |

The technical skills assessment form is a tool for the assessment of knowledge, skills, and the abilities of the participant, compiled by the farm/class room training staff. Like the previous form, respondents could answer by means of a five-point Likert scale (Table 3).

**Table 3.** Indicators of technical skills.

| Indicator | Skill |
|---|---|
| 1 | The individual knows production processes (sowing/transplanting/harvesting...) |
| 2 | The individual knows symptoms related to the presence of pests and parasites |
| 3 | The individual knows techniques and actions related to breeding |
| 4 | The individual knows symptoms related to animal diseases |
| 5 | The individual knows biological cycle (seasonality) of plants/animals |
| 6 | The individual knows soil tillage techniques |
| 7 | The individual knows the use of small tools and equipment |
| 8 | The individual knows the processing techniques and preparation of the products |
| 9 | The individual shows hands on abilities |
| 10 | The individual shows physical ability |
| 11 | The individual is able to adapt to critical situations |
| 12 | The individual is rapid in execution |
| 13 | The individual is precise in carrying out the activities |
| 14 | The individual performs tasks related to production processes |
| 15 | The individual knows how to act in unexpected situations |
| 16 | The individual has ability to manage animals |
| 17 | The individual is able to use tools and equipment |
| 18 | The individual is able to do tool and equipment maintenance |
| 19 | The individual is able to manage working time |
| 20 | The individual is able to work in team |
| 21 | The individual knows how to orientate in the organization |

Soon after the participants started the class room/farm activities, they were asked to fill out the same two forms used by the tutor and the training staff in order to understand the participants' perception of their own skills (self-assessment). This proved to be an effective method to engage participants at a higher level in the assessment process and helped the researcher to elicit a greater depth of response from those participants who may find verbal communication challenging.

In addition, class room/farm training staff and tutors were asked to participate in qualitative in-depth interviews. These interviews, made by the authors, sought to explore the perspectives, views and experiences of the participants' "wider network". The topics discussed included different areas of investigation:

The perceived impact of social farming in general aspects of the participant's life, including any perceived changes in mood, behaviour, or relational capabilities;

The utility of the forms used for the assessment of the technical and psycho-attitudinal skills of the participants;

The strengths and weaknesses of the strategies adopted.

The timeline of data collection, based on the three subjects (participants, tutors, training staff) and two skills assessment instruments (psycho-aptitude and technical), is summarized in Figure 1.

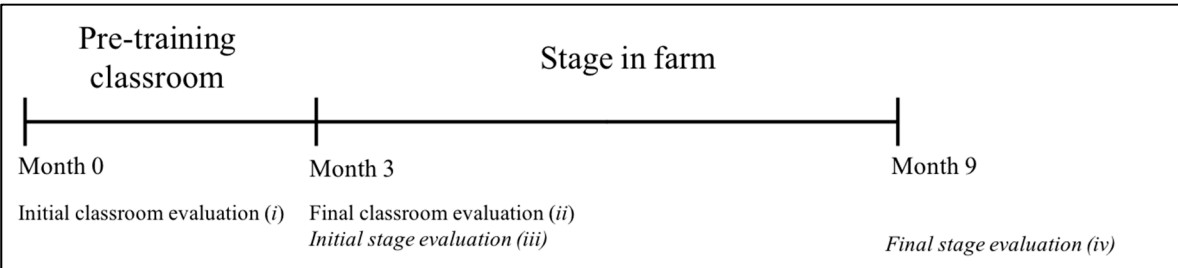

**Figure 1.** Timeline of data collection.

In particular, the psycho-aptitude form (filled by participants and tutors) and the technical form (filled by participants and training staff) were filled out in different periods: (*i*) month zero—initial class room evaluation; (*ii*) month 3—final class room evaluation; (*iii*) month 3—initial stage evaluation; (*iv*) month 9—final stage evaluation. (For more details see Supplementary Materials).

### 3.3. Data Analysis

Nine comparisons between the different assessments provided by participants, tutors, and training staff (using psycho-attitudinal and technical forms) were carried out as follows:

1. Psycho-attitudinal assessment between participants and tutor at period (*i*)
2. Technical assessment between participants and trainer at (*i*)
3. Psycho-attitudinal assessment between participants and tutor at (*iii*)
4. Technical assessment between participants and trainer at (*iii*)
5. Psycho-attitudinal assessment between participants at (*i*) and participants at (*iii*)
6. Psycho-attitudinal assessment between tutor at (*i*) and tutor at (*ii*)
7. Technical assessment between trainer at (*i*) and trainer at (*ii*)
8. Psycho-attitudinal assessment between tutor (*ii*) and tutor at (*iv*)
9. Technical assessment between trainer at (*ii*) and trainer at (*iv*)

These comparisons were useful to understand if any discrepancies were more evident in psycho-attitudinal skills or in technical skills; they were also performed in order to analyze whether the participants had acquired, thanks to the classroom experience, greater awareness regarding their own skills. The data was compiled and organized in an aggregated form by using the average scores reached by each indicator. Missing data was excluded by elaboration.

Statistical analyses were carried out using a non-parametric method, the Chi-squared test: Table 4 shows the comparisons that were statistically significant ($p < 0.005$). The statistical analyses were conducted using Stata for Mac V13.0 (StataCorp LLC, College Station, TX, USA).

**Table 4.** Significant values ($p < 0.005$).

| Comparison | Indicator | Difference Value | Pr |
|:---:|:---:|:---:|:---:|
| 2 | 7 | 0.22 | 0.007 |
| 2 | 8 | 0.00 | 0.007 |
| 2 | 16 | 0.60 | 0.040 |
| 2 | 18 | 0.22 | 0.026 |
| 2 | 20 | 0.00 | 0.011 |
| 2 | 21 | 0.00 | 0.013 |
| 3 | 1 | 0.11 | 0.012 |
| 4 | 2 | −0.29 | 0.030 |
| 4 | 15 | 0.55 | 0.030 |
| 5 | 9 | 0.11 | 0.047 |
| 6 | 1 | 0.00 | 0.030 |
| 6 | 4 | −0.11 | 0.023 |
| 6 | 6 | −0.11 | 0.042 |
| 6 | 9 | 0.00 | 0.030 |
| 6 | 13 | 0.11 | 0.048 |
| 7 | 1 | −1.00 | 0.035 |
| 7 | 7 | −0.33 | 0.026 |

For example, (first row) comparison 2, indicator 7 is referred to "The individual knows the use of small tools and equipment" (see Table 3). It means that the difference between the average value give from participants at (i) (self-assessment) to indicator 7 and the average value given by trainer at (i) to indicator 7 is equal to 0.22. Generally speaking,

- negative differences observed indicate a path that has worsened the competencies of the participants and has negatively modified the aspects connected to the indicators of effectiveness;
- no differences observed; a sign that the course left the participant's skills unchanged and did not affect the aspects related to the effectiveness indicators;
- positive differentials observed indicates a path that has improved the competencies of the participants and has modified the aspects connected to the indicators of effectiveness.

## 4. Results

### 4.1. The Assessment of the Participants

By analyzing Table 4, it is possible to point out some relevant results. For example, Comparison No. 2 (Technical assessment between participants and trainer at (*i*)) was useful to analyze the self-assessment ability of the participant and to understand if this capacity differs based on psycho-attitudinal or technical skills. In the case of the classroom, the initial evaluation of the participant showed to be higher for all the indicators, compared to the initial evaluation made by the tutor/trainer.

However, out of all indicators, only some (those linked to technical skills) showed a variation that was statistically significant: two indicators related to knowledge ("I7 knowledge of the use of small tools and equipment", "I8 knowledge of processing techniques and preparation of the products") and four indicators related to the ability to perform certain tasks ("I16 ability to manage animals", "I18 ability to know how to do tool and equipment maintenance", "I20 ability to work in a team" and " I21 ability to know how to orientate themselves in the organization").

During the farm stage, the initial evaluation made by the participant, which took place in the classroom phase, was higher for all the indicators present in the two assessment forms (except for one indicator).

The only psycho-attitudinal indicator that showed a significant difference (Comparison No. 3—Psycho-attitudinal assessment between participants and tutor at (*iii*)) was "I1 compliance with the rules." This indicator assessed the degree of adequacy with respect to the relationship between the subject and the system of rules, procedures, indications, and clarifications that structure the work experience, and which are necessary for the management of the work and workers and the efficient performance of tasks.

Among the indicators related to technical skills (Comparison No. 4—Technical assessment between participants and trainer at (*iii*)) there were two that showed to be significant; the knowledge of the symptoms related to the presence of weeds and parasites (an indicator for which the evaluation of the participant is lower than that of the trainer) and the ability to know how to act in unexpected situations.

Comparison No. 5 (Psycho-attitudinal assessment between participants at (*i*) and participants at (*iii*)) indicated the capacity of the participants to evaluate their own improvements during the project (from the classroom to the stage on a farm). In terms of psycho-attitudinal skills, a fall in their assessment scores at the beginning of the stage can be observed in comparison to the initial evaluation performed in the class room (a significantly assessed aspect concerning self-care).

Comparisons No. 6 (Psycho-attitudinal assessment between tutor at (*i*) and tutor at (*ii*)) and No. 7 (Technical assessment between trainer at (*i*) and trainer at (*ii*)) showed the results obtained by the participants at the end of the paths and allowed the determination of whether the major improvements were linked to psycho-attitudinal or technical skills. This comparison also aimed to identify the specific aspects that improved as well as the ones that worsened throughout the paths.

By analyzing the data collected on the psycho-aptitude skills survey form (Comparison No. 6), it was possible to highlight some aspects: (*a*) comparing the evaluation provided by the tutor at the beginning and the end of the classroom path showed that there were significant variations for five indicators, three of which were linked to basic skills in work culture ("I1 compliance with the rules", "I4 motivation and interest", "I6 control of the result and ability to solve problems") and two that were

related to psychosocial competencies ("I9 personal care" and "I13 acquisition of organizational skills"); (*b*) for the first three indicators mentioned above, the differences appear to be zero or even positive, indicating a path that left the participants' skills unchanged or even improved them. The situation regarding the other two indicators is worse, particularly in relation to the indicator linked to the acquisition of organizational skills, which is actually diminished; (*c*) examining the differences between the assessments provided by the tutor at the end of the course and at the end of the stage, no significant changes are reported for any of the indicators examined.

Analyzing the data connected with the technical skills scheme (Comparison No. 7) it was possible to observe the following results: (*a*) in the class room, the only indicators for which there was a significant positive variation were "I1 knowledge of production processes" and "I7 knowledge of small tools and equipment"; (*b*) in the farm stage, none of the indicators changed significantly.

### 4.2. The Strengths and Weaknesses of the Paths

Training staff (9 farmers and 1 technical expert for the classroom) and tutors (5 care professionals) were asked through interviews to provide background information about the project and the paths of the participants. All of them agreed that the project was an important collaboration between different subjects (tutor, training staff, counsellor, sending service), which have a common interest in the participant.

> *"The involvement of the training staff, especially inside farms, regarding the observation of participants, has contributed to detecting "signals" and "situations" that otherwise could not have been noticed"* (farmer, interview).

In order to structure time, space and activities, a specific day program was offered at some class rooms and farms. This day program was presented to the participants verbally.

> *"Within it, integrated pauses gave participants the possibility to renovate their energy or, for those who were tired of (group) activities, to retreat and find quietness somewhere on the class room/structure"* (tutor, interview).

This specific (structured) day program did not work the same for everyone and for some participants this was even a problem.

The major concern expressed by some members of the training staff was their lack of knowledge on the participants and their associated behavioral problems. With insufficient knowledge, a farmer may misunderstand participants and may be less able to offer adequate help with their behavioral problems.

> *"The daily presence of the tutor, especially in the class room, facilitated the knowledge of the participants . . . and improved the support provided to the training staff"* (tutor, interview).

In relation to the assessment forms used during the paths, some participants experienced difficulties assessing themselves with respect to some of the indicators (in particular regarding the technical skills assessment form). The general tendency was to use the higher ranking (5 = totally agree). Naturally, this was noticed more significantly in the monitoring of participants belonging to particular categories of disadvantage (for example, mental health).

> *"The indicators in the technical skills assessment form are too specific on certain agricultural activities and clearly do not encompass all the types of activities that may be present at the farm"* (training staff, interview).

At the same time, given that the inclusion in the class room/farm represented a novelty for many of the participants, they tended to give a higher score to almost all the parameters of psycho-aptitude and technical skills than the one provided by the tutor/training staff. This aspect brings attention to the need to further involve the social services of reference in the management of the assignment phases

and the relationships with the tutors/training staff as well as in the activities that have been carried out in this project, which surely represent an aspect to be strengthened and developed further.

Nevertheless, the training staff and tutors felt that the farm stage offered opportunities for participants to be themselves.

> *"They are accepted as people not problems, they are respected, valued, not judged and they are all included and, above all, welcomed"* (tutor, interview).

> *"They perceived that social farming offered a homely, supportive environment where people can experience nature and food production"* (tutor, interview).

The training staff believed that social farming helped participants to connect. The connections that participants could experience included connections with themselves, life, others, food, and nature. In addition, the training staff felt that the care farm offered opportunities for visitors to learn by encouraging them to take part in the activities.

Tutors felt that participants experienced a sense of freedom and security that was developed further throughout the day.

> *"Participants were accepted for who they were on their own terms . . . and they described themselves as feeling happier as a result of having more active social lives"* (tutor, interview).

They were encouraged to ask questions and explore for themselves.

At the same time, the rural locations of the social farming settings meant that participants were often required to travel some distance to attend. For some participants, this motivated them to begin travelling more independently to other settings.

> *"This was also shown to encourage people to travel independently when doing other things, like to go shopping or seeing friends and family"* (farmer, interview).

## 5. Discussion and Conclusions

This study aimed to analyze the psychosocial effects of social farming related to the improvement of personal growth. This scientific research is still at an early stage and often focuses on analyzing the impact of a certain social farming paths on a group of participants to gain an overall view of their effectiveness. Some studies in the literature have provided qualitative analysis (Boyce and Neale 2006; Hassink and Van Dijk 2006; Kawulich 2005; Sempik et al. 2014); our study attempted to provide some results linked to psycho-attitudinal and technical skills.

The analysis of the data collected and the interview material generated pose a rather complicated picture. As argued by Gagliardi et al. (2019) and Sempik (2008), "conducting thorough research in the context of social farming is challenging due to both practical and technical difficulties." The comparison of indicators showed that it was difficult for the participants, mostly those belonging to the mental health area, to evaluate themselves, which was reflected on the differences found between their own assessment and the evaluation made by their tutor and the trainers. This difficulty was greater for technical skills than for psycho-attitudinal skills.

This difference may be due to the fact that the services included participants who, apart from the motivation to enter this type of path, had no previous experience or technical knowledge in agriculture.

The self-assessment ability related to the psycho-attitudinal skills improved at the beginning of the farm stage (compared to the classroom), confirmed by the capacity of participants to understand what was required from them and by having a deeper comprehension of the meaning of the indicators.

The analysis of the assessment forms then suggested that perhaps "too often" the participants were initially scored with the maximum level envisaged by the model and this, in fact, hinders any possibility to improve the initial assessment. Given that their inclusion in the paths represented a novelty for many participants, almost of all the parameters have been assessed by the participants with

a higher rating score than that provided by the trainers. This aspect reflects the need to involve more social services in the management of assigning people to the various paths and in the relationships with the tutors and the activities that have been carried out in this project, but which certainly represents an aspect to be strengthened and developed further.

However, while numerous benefits of social farming were identified, the impact of social farming on the everyday lives of people with intellectual disabilities varied substantially.

The comparison of the indicators showed that the path that brought significant improvements to participants was the classroom phase. In fact, since the classroom is structured with the constant presence of a social tutor, it is easier to follow and monitor the participant's growth path, supporting each individual in dealing in a timely manner with the various difficulties that could arise in the context of work.

As Kaley et al. (2019) highlights, the aspects on which the participants improved are linked to psycho-attitudinal skills, in particular those connected to basic skills in the working world. In addition, the farm stage path is significant, especially in terms of theoretical knowledge of the technical aspects, but not of skills and abilities. In this case, compared to the results from the classroom, coming into contact with a real entrepreneurial reality facilitated the improvement of knowledge in the agricultural field.

As reported previously (Elings and Beerens 2012), farm experiences promote the participants' social relationships with different people, such as tutors, training staff, and other participants sharing the activities. Moreover, in a previous study (Hassink et al. 2010), a specific program of activities in a classroom and on a farm had a positive effect on activity participation, with an increased number of activities performed by the participants.

In spite of all the advantages that could be observed, difficulties in obtaining a completely objective, quick, and comparable measurement of the results achieved by the participants is acknowledged. Among those limitations, it is undoubtedly useful to highlight the complexity of the context (classroom and farm stage) which would require more flexible data collection forms. Also, it would be necessary to train, from the very beginning, one or more individuals to be in charge of specifically monitoring the activities. This would make it possible to obtain greater and richer information (relative to the meaning of the different scores assigned to each aspect of the model), as well as a more holistic understand of the organizational mechanisms and dynamics and the value of the evaluation.

Another limitation of the work could be related to the low number of participants, which was related to difficulties with recruitment (probably due the specific activities and paths). The low number of participants in the project could be justified by the fact that in these types of paths (with people with disabilities), it is important to create groups of people that can work together, but at same time, it is equally important to manage little groups in order to allow tutors and trainers to better follow the participants. In addition, due to a lack of personnel, the farms in which this stage was carried out were willing to take no more than one person with disabilities at a time.

Moreover, some participants dropped out the project before the end of the paths and some did not provide completed forms for the classroom and farm stage. For this reason, the low number of subjects responds to the necessity of analyzing homogeneous data: accordingly, in this work we have used participants who have attended the same environments (in particular, those who have carried out the pre-training activity in the same place).

In general, the evaluation activity has demonstrated the provision of fundamental support in the implementation of the project, playing an active role in supporting the work group involved in the social agriculture supply chain, as well as in the analysis of the activity trend, the identification of points of strength and critical nodes, and in understanding the results achieved.

**Supplementary Materials:** The following are available online at http://www.mdpi.com/2076-0760/8/11/301/s1.

**Author Contributions:** R.M.: Methodology, Writing—original draft, Writing—review & editing. F.D.I.: Conceptualization. A.F.: Data curation, Investigation. P.S.: Investigation. S.E.D.: Writing—review & editing. F.R.: Formal analysis, Writing—review & editing.

**Funding:** This research received no external funding.

**Conflicts of Interest:** The author declares no conflict of interest.

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
