# Peer review of "Social Farming: An Inclusive Environment Conducive to Participant Personal Growth"

_socsci, doi:10.3390/socsci8110301_

Round 1

Reviewer 1 Report

I very much enjoyed reading the paper "Social farming: an inclusive Environment conducive to participant' personal growth" . The paper tries to advocate on the influence of care farming to participants with mental health issues. In so doing it tries to provide a quantification of the data gathered by elaborating on 2 set of indicators (psycho-attitudinal and technical skills) in 9 case studies (people/participants). 

The paper is well written and makes a good argument with good quality in the presentation of a very interesting and popular- both at the academic and public discourse - topic very relevant to the rural development and social economy debate.

Despite all merits i have to point out some thoughts/comments:

i am not so sure that the paper advocates much on the inclusiveness aspect of care farming  (for sure the case study are people with disabilities but this is taken as a fact to move on to the technical skills they aquire). I am not so sure, also, that personal growth can be limited only to the two set of the indicators researched: just a suggestion for the title, i believe it can be changed to actually point out the gist of the study and that is a case study on the methodological tool for skills assessment on a case study social farm in Italy. There is a very important point that this paper must address: terminology . I fully understand that in Italian the word is "social" but to the UK for example setting is translated to care farming. I am not saying it s wrong -on the contrary- the suggestion is to state (in the introduction)  that incosistency  and explain that both terms are used  for the same meaning BUT use only one (and keep it) across al text to my taste i would expect a theoretical framework on tools assessment on educational programmes on the socially excluded and references on tools used. Wy for example we use the self assessment tool instead of an assessment form those next of kin /family/friends or even co-participants? Chapter 3 has no citations. Is it part of the field research? if yes it should be placed elsewhere but if data derive from other previous work then this should be made clear Readers may benefit from some more info on: How many social farms are there in Italy? is "Anche Anche noinoi "a Cooperativa Sociale social cooperative?a GAS? che cosa? When was it established? by how many people? what is the role of the farmer? are the farms male headed? family farms?how big? how close to the participants dwelling?

6.Methodology is quite long chapter (and i totally agree) but when compared to the results there is a mismatch. Hence my proposal to present it as a methodological paper (?)

7. It is not clear what "participant" observation is at this study? maybe it was semi-participant observation? or author's team actually participated during the different  paths ?some other questions have also emerged in this chapter:

how did the indicators (tabel 1 & 2) emerged? from literature?

8. Results section could benefit from a more indepth analysis. I am sure that author/s have lots of field notes that are not well presented - embedded into the discussion. WE have no description on the participants or the people questioned in depth . It is very important ot know who is telling what. Of course not by names but from a general profile.

9. References: the paper can benefit from other work from the Journal social sciences on welfare - social care and other concepts discussed

Author Response

Dear reviewer, we would thank you for your comments and the appreciated suggestions for improving the quality of the manuscript. We made all suggestions in the attached word file. All changes are highlighted in red font in the text. Also, the following responses to you are type in red.

Thanks for your efforts.

Author Response

Dear reviewer, we would thank you for your comments and the appreciated suggestions for improving the quality of the manuscript. We made all suggestions in the attached word file. All changes are highlighted in red font in the text. Also, the following responses to you are type in red.

Thanks for your efforts.

Dr. Francesco Riccioli

Round 2

Reviewer 1 Report

The paper is very much improved and i recommend its publication. The author/s done a great work in addressing all comments.

Author Response

Thanks for your efforts.

Reviewer 2 Report

See attached file.

Author Response

We would thank the Reviewer 2 for the comments and the appreciated suggestions for improving the quality of the manuscript.

All responses to comments are in the attached .docx file.

Round 3

Reviewer 2 Report

See attached file.
